# Estimating spatially disaggregated probability of severe COVID-19 and the impact of handwashing interventions: The case of Zimbabwe

**George Joseph[1], Sveta Milusheva[2], Hugh Sturrock[3], Tonderai Mapako[4], Sophie Ayling[1]\*, Yi Rong Hoo[1]**

1 Water Global Practice, World Bank, Washington, DC, United States of America, 2 Development Impact Evaluation Group, World Bank, Washington, DC, United States of America, 3 Spatial Analysis and Modeling, Locational, London, United Kingdom, 4 Biomedical Research and Training Institute, Harare, Zimbabwe

\* sayling@worldbank.org, sophie2ayling@gmail.com

**Data Availability Statement:** The data sets used in this analysis which are publicly available are the

## Abstract

### Introduction

The severity of COVID-19 disease varies substantially between individuals, with some infections being asymptomatic while others are fatal. Several risk factors have been identified that affect the progression of SARS-CoV-2 to severe COVID-19. They include age, smoking and presence of underlying comorbidities such as respiratory illness, HIV, anemia and obesity. Given that respiratory illness is one such comorbidity and is affected by hand hygiene, it is plausible that improving access to handwashing could lower the risk of severe COVID-19 among a population. In this paper, we estimate the potential impact of improved access to handwashing on the risk of respiratory illness and its knock-on impact on the risk of developing severe COVID-19 disease across Zimbabwe.

### Methods

Spatial generalized additive models were applied to cluster level data from the 2015 Demographic and Health Survey. These models were used to generate continuous (1km resolution) estimates of risk factors for severe COVID-19, including prevalence of major comorbidities (respiratory illness, HIV without viral load suppression, anemia and obesity) and prevalence of smoking, which were aggregated to district level alongside estimates of the proportion of the population under 50 from Worldpop data. The risk of severe COVID-19 was then calculated for each district using published estimates of the relationship between comorbidities, smoking and age (under 50) and severe COVID-19. Two scenarios were then simulated to see how changing access to handwashing facilities could have knock on implications for the prevalence of severe COVID-19 in the population.

### Results

This modeling conducted in this study shows that (1) current risk of severe disease is heterogeneous across the country, due to differences in individual characteristics and

Demographic and Health Survey from 2015 in Zimbabwe (available with registered account) (https://www.dhsprogram.com/methodology/survey/survey-display475.cfm); population density, night-time light intensity for 2016 and distance to nearest Open Street Map (OSM) road obtained from WorldPop (www.worldpop.org). Poverty Small Area Estimations (SMEs) from the World Bank at ward level can be accessed upon request to the World Bank Poverty GP or by writing to one of the World Bank authors, gjoseph@worldbank.org. All of the source code used to generate the datasets are available (https://github.com/dime-worldbank/DiseaseModelling -SSA/tree/main/risk_mapping/ZIM/R). Poverty small area estimation data are available on request to the authors of this report https://openknowledge.worldbank.org/server/api/core/bitstreams/1d1fcadc-43e3-541b-8949-fea45dd2a528/content.

**Funding:** We would like to acknowledge the support of the Global Water Security and Sanitation Partnership (GWSP) of the World Bank and the ieConnect for Impact Program funded with UK aid from the UK government in completing this work. The funders had no role in study design, data collection and analysis, decision to publish, or preparation of the manuscript.

**Competing interests:** The authors have declared that no competing interests exist.

household conditions and (2) that if the quantifiable estimates on the importance of hand-washing for transmission are sound, then improvements in handwashing access could lead to reductions in the risk of severe COVID-19 of up to 16% from the estimated current levels across all districts.

## Conclusions

Taken alongside the likely impact on transmission of SARS-CoV-2 itself, as well as count-less other pathogens, this result adds further support for the expansion of access to hand-washing across the country. It also highlights the spatial differences in risk of severe COVID-19, and thus the opportunity for better planning to focus limited resources in high-risk areas in order to potentially reduce the number of severe cases.

## Introduction

Between January 2020 and March 2023, the Coronavirus pandemic (SARS-CoV-2) affected 220 countries around the world, with over 6.9 million recorded deaths and 677 million recorded cases. Given reductions in tracking and monitoring towards the end of 2022, these numbers were likely higher [1]. In Zimbabwe, there were over 5,000 reported deaths following multiple waves by September 2022 [2].

The severity of COVID-19 disease varies substantially between individuals, with some infections being asymptomatic while others are fatal. Research studies have identified a number of risk factors for severe disease which include older age, smoking and underlying comorbidities including hypertension, diabetes, cardiovascular disease and chronic respiratory disease [3–7]. As these risk factors are themselves spatially variable, the ability to identify whether and where there may be spatially heterogeneous impacts of an outbreak of SARS-CoV-2 on a population could support more effective healthcare policy and planning response scenarios.

Handwashing is widely understood to modify risk of respiratory illness as hands have been known to harbor viral respiratory pathogens [8–10]. This happens when hands come into contact with contaminated fomites (environmental surfaces) or through a person-to-person contact with a disease carrier who sheds the respiratory pathogens through their nose or mouth [11, 12]. Transmission occurs following contact between hands and nasal mucosa or mouth of the new host.

There are at least two channels through which handwashing can theoretically have an impact on reducing the number of severe COVID-19 cases in the population. The more direct pathway is by reducing transmission of the SARS-CoV-2 virus itself. A systematic review by Gozdzielewska et al [13] summarized Health authorities have established that the main transmission routes for SARS-CoV-2 are via either respiratory droplets, inhaled directly by a susceptible individual or through bodily contact [14, 15], or—while not common—via an infected surface or fomite (such as door handles or other frequently touched surfaces) [16–21]. For this reason, handwashing will likely have an impact on transmission of SARS-CoV-2 virus. Handwashing has been encouraged during the pandemic to reduce the likelihood of spreading the infection from one individual to another [17]. There have been more recent efforts to quantify [22] and determine the extent to which both hand-washing facility access [23] and handwashing or hand hygiene behavior [24–27] could impact the transmission of SARS-CoV-2.

However, epidemiological modeling that sought to incorporate handwashing scenarios in predicting disease outcomes in the population at the time had to make assumptions to quantify the impact of handwashing on transmission reduction [28].

The second, more indirect pathway for handwashing in reducing prevalence of severe COVID-19 is via its impact on reducing other respiratory infections. Reducing transmission and incidence of other respiratory illnesses, would reduce prevalence of a risk factor for developing severe COVID-19 [4, 5, 7, 29, 30]. A large body of empirical studies in household and community settings have demonstrated the effectiveness of hand washing in reducing other respiratory illnesses around the world, be that through campaigns or through the presence of the facility itself. Washing hands with soap (both plain and antibacterial soap) is a proven mechanism to eliminate bacteria and respiratory viruses [31–33]. A systematic review of eight studies from a pool of 410 articles found that hand washing lowered the risk of respiratory infection, with risk reductions ranging from 6% to 44%, but noted that a greater number of rigorous studies are urgently needed [34]. Another meta-analysis suggested that regular hand hygiene provided a significant protective effect (OR = 0.62;95% CI 0.52–0.73; I2 = 0%) against 2009 pandemic influenza infection [35]. A randomized controlled trial among squatter settlements in Karachi, Pakistan, found that children younger than 5 years old living in households that were assigned to antibacterial soap had roughly a 50 percent lower incidence of pneumonia than controls [36]. Similarly, another randomized control trial across 60 elementary schools in Egypt found that a hand hygiene intervention campaign significantly reduced the incidence of laboratory-confirmed influenza by 50 percent and subsequently absenteeism among the students caused by the illnesses [37]. In Bangladesh, a study over 60,000 low-income households, found that across the sample, respiratory illness was lower among people who had soap and water present in the hand washing station than among those who did not [38]. A two-stage cluster RCT with 60 villages in Kenya also showed that children in schools where hand washing and drinking water stations were made available were consistently less likely to have an acute respiratory infection over the following 2–10 months than in schools without (2% versus 3%, Estimated Difference in Medians -2%, 90% CI-3% to -1% in rounds 1–17) [39].

Taken together, it is therefore plausible that handwashing could impact the number of hospitalizations and deaths from COVID via two pathways: 1) by reducing transmission and incidence of COVID-19 and 2) by reducing the incidence of underlying respiratory illness in a population, thereby reducing susceptibility to severe COVID-19. If a host who becomes infected with SARS-CoV-2 already has a respiratory infection the chances of that developing to severe COVID-19 disease are increased. What we focus on in the analysis presented in this paper is therefore the second, more indirect pathway described above, to impact on prevalence of severe COVID-19 cases.

This study contributes to the literature in three main ways. Firstly, it contributes to the growing body of spatially disaggregated approaches to COVID-19 but focusing on clinical risk data. Spatially disaggregated models have made use of mobility data to examine transmission dynamics [40, 41] and examined the impact of government policies [42], through lockdowns, or compared population characteristics in their response to government measures [43, 44]. However, to the best of our knowledge, this study is one of the few, particularly in the sub-Saharan African context, that have used available spatially disaggregated data on clinical risk factors to map underlying risk of severe COVID-19 in a population. Understanding spatial variation in risk using spatial modeling has proved useful for a number of diseases including malaria [45], schistosomiasis [46], and HIV [47]. In turn, clinical risk for severe COVID-19 is likely to differ across a country, so using a spatial approach has potential to be strategically

useful for targeting a health policy response. Its application to severe COVID-19 risk factors is therefore applied here.

Secondly, though it is widely accepted that hand hygiene is important for reducing the spread of respiratory illnesses, and that it is important for reducing transmission of SARS--CoV-2, there have been limited efforts to map out the pathways for how handwashing can impact upon a reduction in severe COVID-19 cases. This is despite the use of hand hygiene scenarios that reduce transmission being incorporated into epidemiological models [28]. In this study we seek to build on existing literature that sought to quantify the impact of hand-washing on reducing the transmission of respiratory pathogens and examine how that could have knock-on implications for the extent of severe COVID-19 prevalence.

Thirdly, this paper relies on available data to map underlying risk factors of severe COVID-19 disease and potential impact of handwashing interventions in a way that can be used for guiding policy and planning interventions. This approach is particularly useful in data scarce environments, common to developing countries.

## Data and methods

### Study area

In this paper we focus on Zimbabwe as a case study for the methodology being presented. Though there is an emerging collection of COVID-19 related studies within the African continent [48], the majority of studies thus far have been concentrated in Asia, North America, and Europe, while Africa focused studies are in the minority [49]. With lower rates of vaccination on average than in other parts of the world, vulnerability to severe disease remains an important concern in countries like Zimbabwe. Spatial analysis methods of this nature can be taken as a useful model for similar contexts where vaccine prioritization may be important in public health planning. It is also a data scarce environment when compared to some of the countries studied more intensively during the pandemic, and thus can provide a useful example of how similar contexts can be studied under such circumstances.

### Data

The main data sets used in this analysis are: the Demographic and Health Survey (DHS) from 2015 in Zimbabwe; population density, night-time light intensity for 2015 and distance to nearest Open Street Map (OSM) road obtained via WorldPop (www.worldpop.org); and poverty Small Area Estimations (SMEs) from the World Bank at ward level [50]. The datasets used are fully outlined in Table 1.

The 2015 DHS was used to inform the creation of a hand-washing risk index drawing from data on access to hand-washing facilities with soap and water at the household level. Respiratory illness data was taken from the same DHS survey but only available for children under five years of age. Though further data was due to be collected in 2020, this has yet to be released due to delays during the pandemic. Studies have indicated that children can often be the source of respiratory illness transmission to adults within the same household [51, 52], and thus we have used this indicator as a proxy for respiratory illness prevalence in adults at the household level. Anemia, overweight prevalence, and HIV prevalence were also taken from this source.

There are 400 clusters in the DHS distributed across the ten provinces (60 districts) of the country. Though the data is originally collected to be representative at the province level, the application of spatial modeling to such data allow for higher resolution predictions of outcomes [53, 54] such as malaria prevalence or vaccine coverage e.g., [55, 56]. Though DHS clusters are intentionally displaced by 0-2km in urban areas and 0–5 km in rural clusters, with a further, randomly selected 1% (every 100th) of rural clusters displaced by up to 10km [57].

**Table 1. Data sources used to generate the covariates in the analysis.**

| Data Description | Data Type, Resolution, Dates, Obs, sampling technique | Source and Availability |
|---|---|---|
| Cluster locations and associated household characteristics | De-identified GPS coordinates of household respondents, handwashing facilities in the home, respiratory illness prevalence, anemia, overweight prevalence, and HIV prevalence | DHS 2016 https://dhsprogram.com/data/available-datasets.cfm |
| **Pre-existing geospatial grids** | | |
| Population density data | Raster | Facebook HDX Humanitarian Data Exchange https://data.humdata.org/dataset/worldpop-population-density-for-zimbabwe |
| Night-time light brightness and Digital Elevation Model | Global dataset Resolution | NASA (2015) 'ASTER Level 1 Precision Terrain Corrected Registered At-Sensor Radiance Version 3'. Sioux Falls, South Dakota: NASA EOSDIS Land Processes DAAC, USGS Earth Resources Observation and Science (EROS) Center. |
| | | https://www.earthdata.nasa.gov/learn/backgrounders/nighttime-lights |
| | | https://solarsystem.nasa.gov/resources/585/asters-global-digital-elevation-model/ |
| Distance to the nearest Open Street Map Road | Raster | Open StreetMap https://www.openstreetmap.org/#map=5/51.330/10.453 |
| Poverty Small Area Estimates | Polygon shapefile | World Bank Poverty Global Practice https://microdata.worldbank.org/index.php/home |

The application of interpolation techniques specifically to DHS data is described in the referenced publication [54]. In addition, several other studies have also implemented the same interpolation techniques using DHS data [56, 58–60]. Spatial modeling approaches leveraged are described in more detail in the methods section.

The odds ratios for risk of severe disease from different clinical risk factors were taken from a meta-analysis and systematic review study of incidence and clinical characteristics and prognostic factors of patients with COVID-19 across 53,000 patients [61] and the age specific probabilities of hospitalization are taken from *Estimates of the severity of coronavirus disease 2019: a model-based analysis* the estimates of which can be found in S1 Table [62]. All of the source code used to generate the datasets are available at https://github.com/dime-worldbank/Disease-Modelling-SSA/tree/main/archive/risk_mapping/ZIM/R.

## Methods

The methodology outlined here includes three parts. First, we calculate the prevalence of risk factors and comorbidities for severe COVID-19 at the district level, drawing on the interpolated data. Comorbidities include prevalence of respiratory illness modeled as a function of handwashing risk, anemia and obesity. Additional risk factors included prevalence of smoking and fraction of the population under the age of 50. To generate district level estimates, we applied spatial Generalized Additive Models to georeferenced cluster level DHS data to generate continuous (1km grid cells) estimates of prevalence of risk factors which are then aggregated to district level. Secondly, we use these district level prevalence estimates of risk factors to calculate the probability of developing severe COVID-19 disease using published estimates of effect sizes for these risk factors. Thirdly, we modify the level of handwashing risk according to different counterfactual scenarios in order to see the impact it has on reducing prevalence of respiratory illness and severe COVID-19 risk across the country. (All of the following analysis were conducted in R).

**Predicting prevalence of risk factors and comorbidities.** *Handwashing risk*. Using the three variables related to handwashing in the DHS data, we assign each household a hand

washing 'risk' score. For households with a hand washing facility with water and soap, this was considered equivalent to a hand washing risk of 0. With water but no soap, there was a risk of 0.5, while with no water or no facility (even if soap was present) the risk was set to 1. The mean household level observation was then calculated for each cluster.

Cluster level georeferenced handwashing risk scores were used to predict handwashing risk across the whole country and generate a continuous map using a beta Generalized Additive Model (GAM) with a logit link function. Population density, night-time light intensity for 2015 and distance to nearest Open Street Map (OSM) road obtained via WorldPop (www. worldpop.org) were all resampled to 1km resolution and included as covariates with thin-plate splines used to model non-linear effects. These covariates were chosen due to their plausible association with poverty and access to handwashing facilities. While other covariates could be used, here the goal of the spatial modeling was to produce predictions of handwashing risk as opposed to examining factors associated with handwashing risk in detail. We also included a bivariate smooth on latitude and longitude to account for spatial autocorrelation and additional penalty term in the model to allow covariates that were not related to the outcome to essentially be removed from the model [63]. This model was used to predict handwashing risk at 1km resolution. District level handwashing risk was then estimated by calculating the population weighted mean for all cells within a district.

*Respiratory illness*. To generate continuous maps of respiratory illness, we calculated the prevalence of respiratory illness symptoms at any time in the two weeks preceding the survey at each DHS cluster, available for individuals aged under 5 years of age [64]. The prevalence of respiratory illness symptoms was then modeled using a binomial GAM with handwashing risk, distance to road, population density and poverty as covariates in addition to a spatial bivariate smooth. The ward level poverty estimates were rasterized to be on the same 1km resolution grid as all other covariates and included as an additional covariate. This model was then used to predict prevalence of respiratory illness at 1km resolution. District level prevalence of respiratory illness was then estimated by calculating the population weighted mean for all cells within a district.

*Comorbidities and factors for the probability of severe COVID-19*. In addition to respiratory illness, we generated estimates of the prevalence of several additional comorbidities and probability factors for severe COVID-19 at the district level. Individual probability factors that are included at this stage are based on existing literature and presently are HIV prevalence (without Viral Load Suppression), anemia prevalence, and obesity prevalence, in addition to respiratory illness. For each of these, prevalence was calculated at the cluster level and the same GAM model was fit to each outcome to generate predicted values at the 1km cell level. Once these gridded surfaces are created at a 1km resolution, estimates are reached at the district level by weighting, using a raster file of population count at the pixel level.

We then calculate the probability of having any of these comorbidities, defined as 1 minus the probability of having none of these comorbidities:

$$C = 1 - (1 - (R.H.A.O))$$

where *C* = proportion with a comorbidity, *R* = proportion with respiratory illness, *H* = proportion with HIV (without viral load suppression), *A* = proportion with anaemia and *O* = proportion with obesity.

We apply the same geospatial methods to calculate the proportion of the population that is a smoker based on smoking prevalence within DHS clusters. We additionally calculate the proportion of the population that is over age 50 from WorldPop data.

**Predicting probability of severe COVID-19.** To calculate the probability of severe COVID-19, we apply the odds ratios estimated in Ma et al to the proportion with the corresponding probability factor/comorbidity at the district level.

$$logit(SC) = \log(2.6).C + \log(1.7).S + \log(2.6).p50 + intercept$$

where *SC* = probability of severe COVID-19, *C* = proportion with a comorbidity, *S* = proportion that smoke, *p50* = proportion over 50.

In order to apply this equation, we need the intercept value specific to Zimbabwe. This was set so that the mean district level probability of severe COVID was equal to that expected on the basis of the age-breakdown of the population. To estimate this expected probability of severe COVID, we applied the Imperial College London (ICL) clinical study age-specific estimates of probability of severe disease to the age breakdown data from the Zimbabwe census from 2012. This generated a mean probability of severe disease across the population in Zimbabwe of 2.9%. After testing a range of different intercepts, an intercept of -4.2 was chosen as this produced a mean district level probability of severe disease of 2.9%.

**Generating counterfactual scenarios.** We produce two counterfactual scenarios for probability of severe COVID-19. These are based on predictions of prevalence of respiratory illness under different levels of handwashing risk, which affect the probability of having a comorbidity and in turn the final probability score. The first scenario uses takes the existing heterogenous values of handwashing risk throughout the country and adjusts them each to 0.25 throughout the country, equivalent to the lowest level of handwashing risk (i.e., the highest level of handwashing access) currently observed at the district level. This is a realistic scenario where it is assumed that all the districts can improve their handwashing access levels *at least* to the level of the best performing district. The second scenario, which shows an ideal case, uses a handwashing risk of 0, equivalent to assuming that *every* household has access to a hand washing facility, soap and water.

**Ethics statement.** This study was approved by the Medical Research Council of Zimbabwe. The approval number is MRCZ/E/302 and it was assessed as exempt. All data obtained and herein reported were anonymized prior to analysis.

## Results

### Handwashing risk estimates

Fig 1 shows the cluster level handwashing risk, and the district level estimates. There is considerable variation across Zimbabwe, with the lowest cluster level handwashing risk of 0.036 and highest of 0.923. At the district level, hand washing risk varied from 0.25 in Bulawayo to 0.69 in Mberengwa.

### Respiratory illness probability estimates

Fig 2 shows the relationships between respiratory illness and each of the covariates explored, including the spatial effect. As expected, handwashing risk showed a general positive relationship with respiratory illness as did population density. There appeared to be a negative, but non-linear, relationship with distance to road. Poverty did not show an association with respiratory illness. In addition to the covariate effects, there appeared to be residual spatial variation in respiratory illness across the country not explained by those covariates included in the model.

Predicted prevalence of respiratory illness varied across the country, with Makonde, Gokwe North and Kariba districts all having a predicted prevalence of > 0.54 and Insiza, Beitbridge and Matobo having a prevalence of < 0.26 (Fig 3).

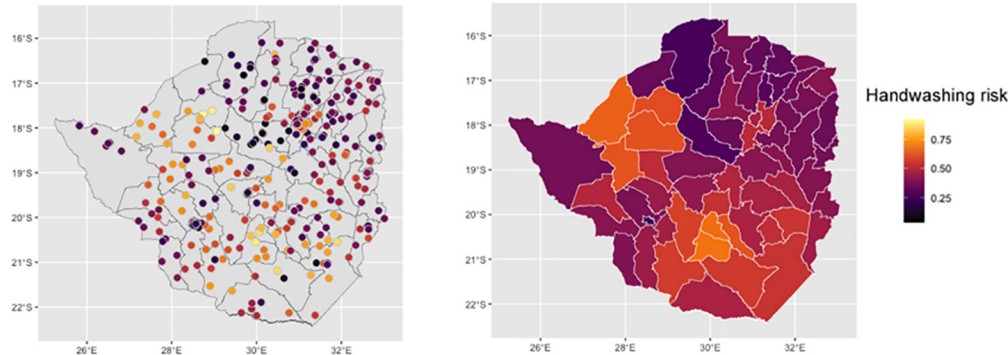

**Fig 1.** DHS Cluster level handwashing risk across Zimbabwe in 2015 (left) and estimated district level handwashing risk (right). Basemap attribution: OpenStreetMap contributors https://www.openstreetmap.org/.

## Estimates of probability of severe COVID-19

Estimates of the probability of severe disease among symptomatic cases are displayed in Fig 5. Under current conditions, probability of severe disease among symptomatic cases at the district level is predicted to vary between 0.026–0.033, with Makonde, Mutasa and Marondera districts predicted to have the highest levels of probability (Fig 4).

**Counterfactual scenarios.** Under the first counterfactual scenario of reducing handwashing risk to the lowest observed levels at district level, mean prevalence of respiratory illness is predicted to drop from 0.39 to 0.32 (Fig 5B). Under the second scenario of perfect handwashing access, it drops to 0.15 (Fig 5C).

This decrease in the prevalence of respiratory illness leads to a drop in the probability of severe disease (Fig 6B and 6C). This drop is marginal if handwashing risk is set to 0.25

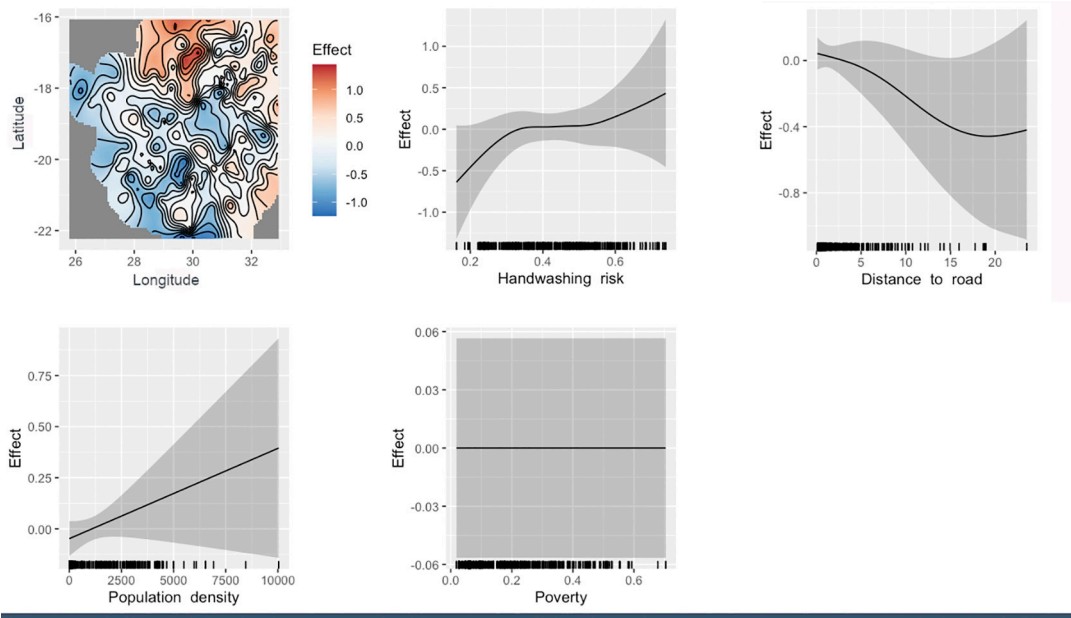

**Fig 2. The estimated associations between the risk factors explored and prevalence of respiratory illness in Zimbabwe, including the spatial effect (lng, lat).** Effect is on the log odds scale.

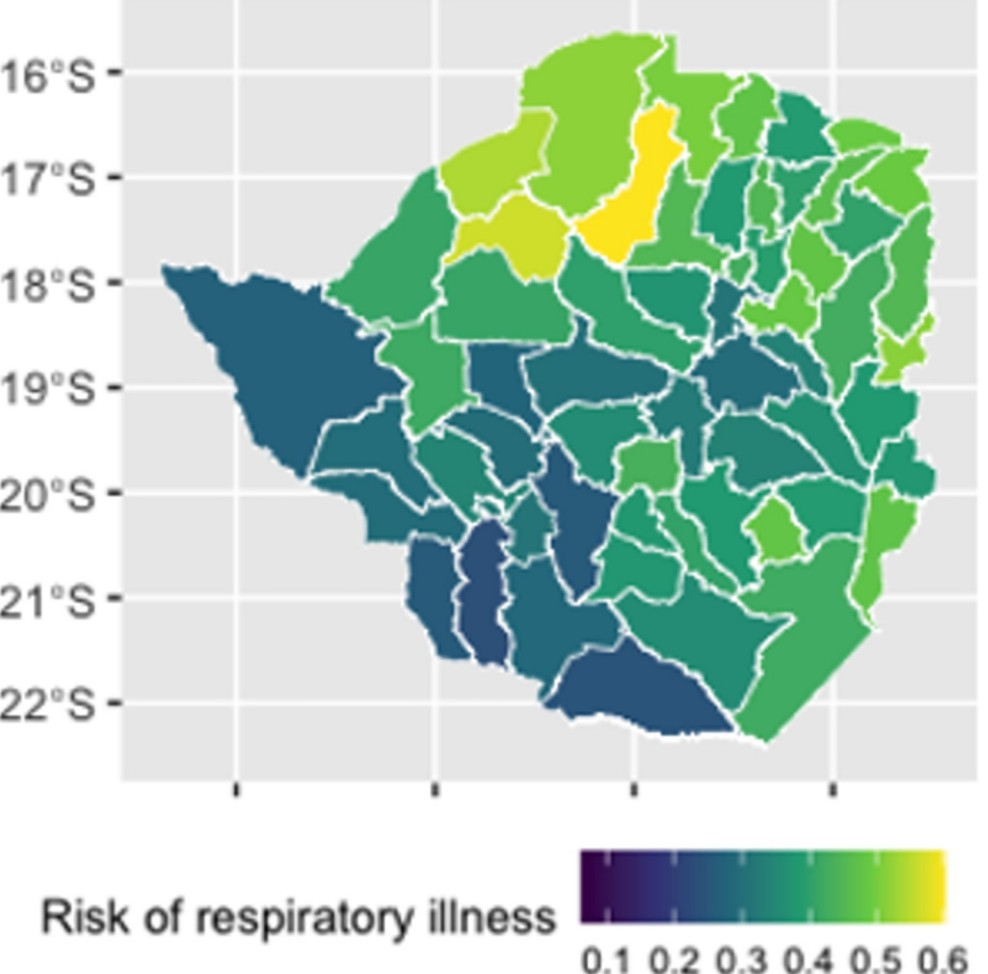

**Fig 3. Predicted prevalence of respiratory illness at the district level in Zimbabwe.** Basemap attribution:
OpenStreetMap contributors https://www.openstreetmap.org/.

throughout every district (Fig 6B), with the mean district level probability of severe disease
dropping from 0.029 to 0.028. This drop in risk is more substantial in a scenario of perfect
access to handwashing facilities (i.e., a handwashing risk of 0), with the mean district level
probability of severe disease dropping by around 16% to 0.025 (Fig 6C).

## Discussion

The severity of COVID-19 disease caused by SARS-CoV-2 is highly variable with some indi-
viduals and populations being more vulnerable to complications than others [65, 66]. Some
other studies have identified spatial variation in COVID-19 severity associated with tertiary
workers being at higher risk in India [67], areas with greater socio-economic inequality in
Spain [68] and lower incomes and high poverty rates in the US [69]. However, few studies, par-
ticularly in the sub-Saharan African context have used available spatially disaggregated data on
clinical risk factors to map underlying risk of severe COVID-19 in a population [70]. Using
information on known factors for probability of severe disease and mapping it to available

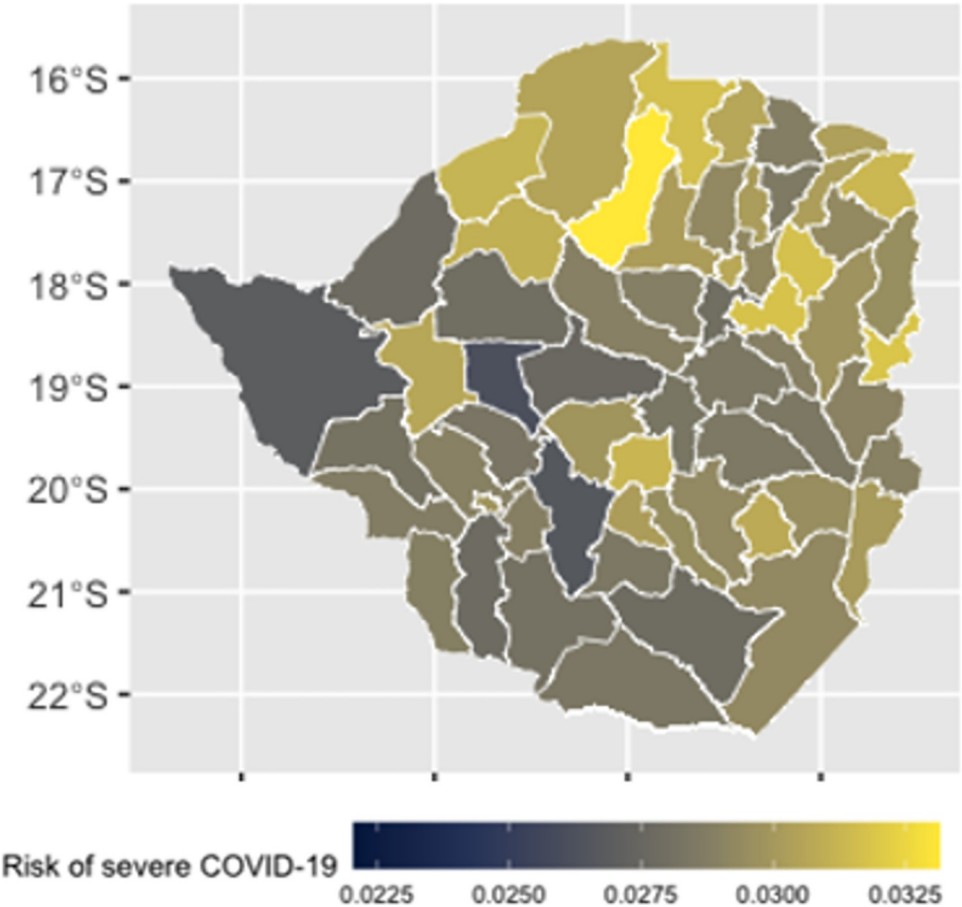

**Fig 4. Predicted prevalence of severe COVID-19 at the district level in Zimbabwe.** Basemap attribution: OpenStreetMap contributors https://www.openstreetmap.org/.

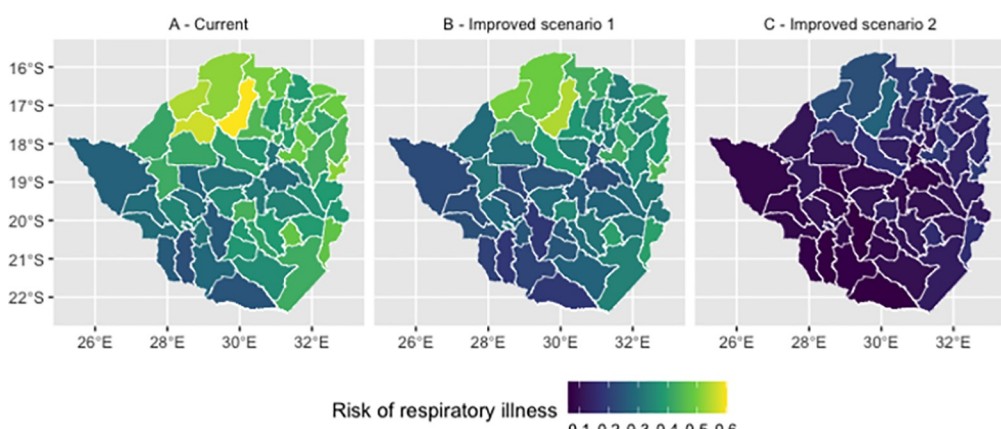

**Fig 5. Predicted prevalence of respiratory illness at the district level in Zimbabwe—current, and under improved handwashing scenarios 1 and 2.** Basemap attribution: OpenStreetMap contributors https://www.openstreetmap.org/.

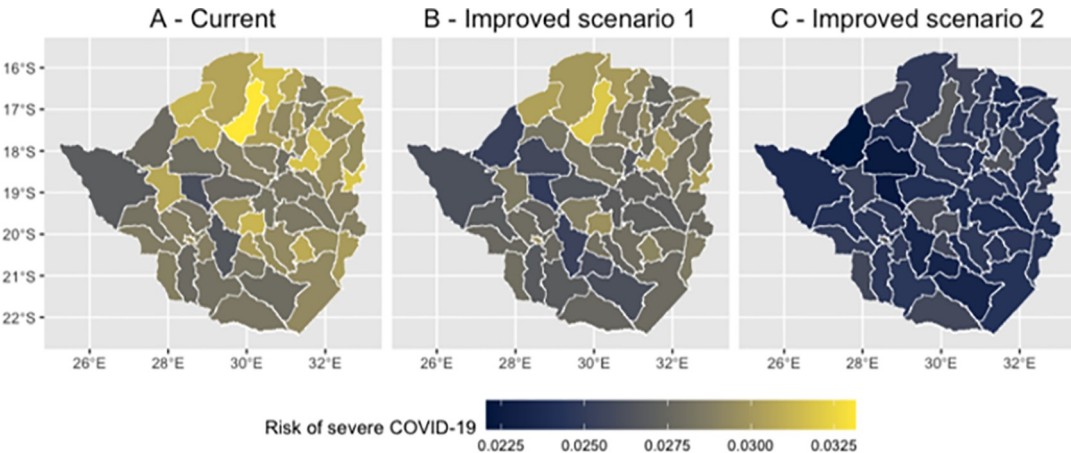

**Fig 6. Predicted probability of severe COVID-19 among symptomatic cases at the district level in Zimbabwe under different hand washing risk scenarios 1 and 2.** Basemap attribution: OpenStreetMap contributors https://www.openstreetmap.org/.

data on such factors from the DHS, here we estimate and map this variation using GAM [71, 72] in probability of severe disease across Zimbabwe. Results show that the probability of severe disease varies substantially with some districts having a probability of severe disease nearly 30% higher than others. Furthermore, we show that there is a relationship between handwashing and respiratory illness, a potential risk factor for severe disease, and that improving access to handwashing across the country could reduce the probability of severe disease.

From a policy perspective, this study highlights a potential added benefit of improvements to WASH facilities in addition to the already known numerous other benefits improved WASH brings. Hand washing is also an important intervention for gastro-intestinal and worm infections [34, 73–76], in addition to the relationships highlighted here for reducing transmission of respiratory infections. Taken together, this study adds another perspective on why handwashing access, particularly in developing countries, can continue to be considered as an important tool in the fight against COVID-19. Other studies have also shown it to be highly cost-effective in prevention of the spread of the virus [77], an important consideration in light of the considerable economic and societal costs of the pandemic, and associated restrictions [78, 79]. Indeed, investments in WASH are likely to lead to substantial economic returns in the absence of COVID-19. The pandemic should make this investment all the more urgent.

## Limitations

While the link between handwashing and probability of severe COVID-19 is plausible, there are a number of important limitations and assumptions made as part of this study that deserve discussion. Firstly, we assumed that the prevalence of respiratory illness symptoms in children is representative of the prevalence of illness in the wider population. In reality, rates of symptoms and disease may vary between age-groups. Furthermore, many types of respiratory illness may not be severe enough to constitute being a risk factor for severe COVID-19.

Second, our results rely on the strong relationship between handwashing and respiratory illness risk based on previous studies, the causality of which has been indicated in the literature reviewed in the introduction [9, 11, 36]. However, this model does not take into account additional factors affecting respiratory illness risk e.g., indoor pollution [80, 81]. Further to this point, out of necessity, these analyses are conducted at district level using odds ratios obtained

from analysis at the individual level which is not representative of the full population. While we don't expect this to impact the main conclusions of the study, it raises the possibility of the atomistic fallacy.

Third, the comorbidities and risk factors for severe disease included in this study are not exhaustive and are limited to those for which data are available. Other factors that may pose a risk for severe disease, include TB, malaria, or hypertension (or yet to be discovered others). Further to this point, we only consider prevalence of *any* comorbidity as a risk factor. In reality, having more than one comorbidity is likely to increase the probability of severe disease. To incorporate this into this study would require estimates of odds ratios for each risk factor and their interactions as well as an understanding of the correlation between risk factors as these are likely to not be independent. For example, obesity has been linked to hypertension [82]. What the model seeks to do is provide a framework into which more data on risk factors can be added as and when it becomes available.

Fourth, we make the assumption that the impact of improvements in access to handwashing on prevalence of respiratory illness occurs over a timescale relevant to the pandemic. While previous research from Kenya suggests that the impact can be observed 2–9 months after intervention [39] if this timescale is longer in Zimbabwe, or if the improvements themselves take a long time to be implemented, we may not see the desired effects. One potential way to maximize impact would be to prioritize districts for improved handwashing according to their current handwashing risk. Intervention impacts could be maximized by targeting districts or communities with highest handwashing risk first. Outputs from ongoing spatially discrete agent-based modeling, informed by human movement data, could also be used to identify those districts most vulnerable to transmission.

Fifth, there are also limitations to the data that was used. The analysis was conducted during the time of emergency response so there was still limited information available on risk factors. The DHS data that was used is from 2015 due to the DHS 2020 data collection having been paused due to the pandemic. In 2023 this dataset is still not available. However, the analysis could be re-run in future with updated datasets.

Finally, this paper has only focused on underlying risk factors in the population and has not considered risk of transmission due to e.g., population mobility. The outputs of this work will likely therefore be used as part of a larger agent-based modeling exercise to identify those districts with highest probability of severe disease when transmission is also taken into account, in a separate, forthcoming paper.

In conclusion, this study shows the spatial disparity in severe COVID-19 risk across Zimbabwe. Secondly, it shows that there is a relationship between access to handwashing facilities and respiratory illness, a known risk factor for severe COVID-19. In addition to any direct effects on transmission of SARS-CoV-2, improving access to handwashing may therefore reduce the probability of severe disease among the population which could in turn reduce the number of individuals requiring hospitalization, critical care or dying from the disease.

## Supporting information

**S1 Table. Assumed hospitalization rates for symptomatic infections by age category.** (DOCX)

## Acknowledgments

This analysis formed part of a broader collaboration of the World Bank with the Zimbabwe National Modeling Consortium. In particular the Biomedical Research and Training Institute

—Institutional Review Board (BRTI-IRB) and the COVID-19 Modeling and Spatial analysis sub-group (modeling consortium); Mr. Tendayi Kureya, Executive Secretary of the Medical Research Council of Zimbabwe (MRC-Z); Dr Shungu Munyati, Director General of BRTI; Professor in Demography and Behavioral Science Simon Gregson and Dr. Mike Pickles, Research Fellow of Imperial College London (ICL). From the World Bank we would like to thank Chenjerai Sismayi from the Health, Nutrition and Population Global Practice of the World Bank as well as the Country Management Unit of the World Bank in Zimbabwe.

The findings, interpretations and conclusions expressed in this paper do not necessarily reflect the views of the World Bank, the Executive Directors of the World Bank, or the governments whom they represent. The World Bank does not guarantee the accuracy of the data included in this work.

## Author Contributions

**Conceptualization:** George Joseph, Sophie Ayling, Yi Rong Hoo.

**Data curation:** Sveta Milusheva, Hugh Sturrock, Tonderai Mapako, Sophie Ayling, Yi Rong Hoo.

**Formal analysis:** Sveta Milusheva, Hugh Sturrock, Yi Rong Hoo.

**Funding acquisition:** George Joseph.

**Investigation:** Sophie Ayling.

**Methodology:** George Joseph, Sveta Milusheva, Hugh Sturrock, Tonderai Mapako.

**Project administration:** George Joseph, Sophie Ayling.

**Supervision:** George Joseph, Sveta Milusheva.

**Validation:** Hugh Sturrock, Tonderai Mapako.

**Visualization:** Hugh Sturrock.

**Writing – original draft:** Sveta Milusheva, Hugh Sturrock.

**Writing – review & editing:** George Joseph, Hugh Sturrock, Tonderai Mapako, Sophie Ayling, Yi Rong Hoo.

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
