## [Decision Letter · Decision Letter 0]

13 Feb 2023

PONE-D-22-34494Estimating spatially disaggregated probability of severe COVID-19 and the impact of handwashing interventions: The case of ZimbabwePLOS ONE

Dear Dr. Ayling,

Thank you for submitting your manuscript to PLOS ONE. After careful consideration, we feel that it has merit but does not fully meet PLOS ONE’s publication criteria as it currently stands. Therefore, we invite you to submit a revised version of the manuscript that addresses the points raised during the review process.

Please submit your revised manuscript by Mar 30 2023 11:59PM.  If you will need more time than this to complete your revisions, please reply to this message or contact the journal office at plosone@plos.org. Please include the following items when submitting your revised manuscript:A rebuttal letter that responds to each point raised by the academic editor and reviewer(s). You should upload this letter as a separate file labeled 'Response to Reviewers'.A marked-up copy of your manuscript that highlights changes made to the original version. You should upload this as a separate file labeled 'Revised Manuscript with Track Changes'.An unmarked version of your revised paper without tracked changes. You should upload this as a separate file labeled 'Manuscript'.If applicable, we recommend that you deposit your laboratory protocols in protocols.io to enhance the reproducibility of your results. Protocols.io assigns your protocol its own identifier (DOI) so that it can be cited independently in the future. For instructions see: https://journals.plos.org/plosone/s/submission-guidelines#loc-laboratory-protocols. Additionally, PLOS ONE offers an option for publishing peer-reviewed Lab Protocol articles, which describe protocols hosted on protocols.io. Read more information on sharing protocols at https://plos.org/protocols?utm_medium=editorial-email&utm_source=authorletters&utm_campaign=protocols.

We look forward to receiving your revised manuscript.

Kind regards,

Bedilu Alamirie Ejigu, Ph.D

Academic Editor

PLOS ONE

Journal Requirements:

"We would like to acknowledge the support of the Global Water Security and Sanitation Partnership (GWSP) of the World Bank and the ieConnect for Impact Program  funded with UK aid from the UK government in completing this work. "

"NO authors have competing interests"

7. We note that Figures 1, 3, 4, 5 and 6 in your submission contain [map/satellite] images which may be copyrighted. All PLOS content is published under the Creative Commons Attribution License (CC BY 4.0), which means that the manuscript, images, and Supporting Information files will be freely available online, and any third party is permitted to access, download, copy, distribute, and use these materials in any way, even commercially, with proper attribution. For these reasons, we cannot publish previously copyrighted maps or satellite images created using proprietary data, such as Google software (Google Maps, Street View, and Earth). For more information, see our copyright guidelines: http://journals.plos.org/plosone/s/licenses-and-copyright.

    1. You may seek permission from the original copyright holder of Figures 1, 3, 4, 5 and 6 to publish the content specifically under the CC BY 4.0 license. 

Please upload the completed Content Permission Form or other proof of granted permissions as an "Other" file with your submission”

Additional Editor Comments:

Dear Ms. Sophie Ayling,

I have completed my evaluation of your manuscript. The reviewers recommend reconsideration of your manuscript following carful revision. I invite you to resubmit your manuscript after addressing the comments below. Please resubmit your revised manuscript by March 15, 2023.

Dealing with the estimation of spatially disaggregated probability of COVID-19 infection is an important research topic. Here the authors report the impact of improved access to handwashing on the risk of respiratory illness with spatial focus on COVID-19.

1) Abstract

- Please split this into introduction, methods, results and conclusions

2) Introduction

- This is a very good background, but the following should be addressed

i) Update the figures in the first paragraph using recent number

ii) some paragraphs lack references, such as the fourth paragraph

iii) clearly state the research problem

3) Data and Methods

- Move the “Ethics Statement” at the end of this Section, i.e. before the Result Section

- Line #131, don’t you think DHS2015 is an outdated data?

- Line #147, how the intentionally displaced DHS cluster location taken into account in the interpolation process?

- Line #154, the provided Github link not working.

- Line #158, are your sample representative to perform district level estimation?

- Line #178-180, you state about the spatial model, but unable to its clear description. How the model stated/formulated?

- Line #222, the counterfactual scenarios are similar across the country. Since there is a considerable variation in handwashing practice in Zimbabwe (see your statement line #233-244), how your assumed scenarios will be valued?

4) Results

- Fig2 (top-left), write the correct label for x- and y-axis

- Line #242. Handwashing risk highly linked with household wealth. How the authors justify the non-significant link between poverty and respiratory illness?

5) Discussion

- Lacking to compare/citation with other studies, i.e., line #301

- Why no limitations discussion here? Please add a Limitation section.

Reviewers' comments:

Reviewer's Responses to Questions

**Comments to the Author**

1. Is the manuscript technically sound, and do the data support the conclusions?

Reviewer #1: Yes

Reviewer #2: Yes

2. Has the statistical analysis been performed appropriately and rigorously? 

Reviewer #1: Yes

Reviewer #2: Yes

3. Have the authors made all data underlying the findings in their manuscript fully available?

Reviewer #1: Yes

Reviewer #2: Yes

4. Is the manuscript presented in an intelligible fashion and written in standard English?

Reviewer #1: Yes

Reviewer #2: Yes

5. Review Comments to the Author

Reviewer #1: The paper discuss the severity of COVID-19 disease in different parts of Zimbabwe The disease varies substantially between individuals, with some are asymptomatic while others are fatal. Several risk factors have been identified that affect the progression of SARS-CoV-2 to severe COVID-19. To mention some age, smoking and presence of underlying comorbidities such as respiratory illness, HIV, anemia and obesity. Given that respiratory illness is one such comorbidity and is affected by hand hygiene, it suggests that improving access to handwashing could lower the risk of severe COVID-19 among a population. In the paper, the potential impact of improved access to handwashing on the risk of respiratory illness was estimated and handwashing knock-on impact on the risk of developing severe COVID-19 disease across Zimbabwe was estimated. they have used a geospatial model that allows to estimate differential clinical risk at the district level.

Results:- shows that the current risk of severe disease is heterogeneous across the country, due to differences in individual

characteristics and household conditions.

- demonstrates how household level improved access to handwashing could lead to reductions in the risk of severe COVID-19 of up to 16% from the estimated current levels across all districts. Taken alongside the likely impact on transmission of SARS-CoV-2 itself, as well as countless other pathogens,

this adds further support for the expansion of access to handwashing across the country.

It also highlights the spatial differences in risk of severe COVID-19, and concluded that it shows the opportunity for better planning to focus limited resources in high-risk areas in order to potentially reduce the number of severe cases.

my comments are:-

1)The DHS used is from 2015 while they could have used a recent one published on 2020.

2) The captions used for the figures are not well descriptive.

Reviewer #2: Manuscript Number: PONE-D-22-34494

Full Title: Estimating spatially disaggregated probability of severe COVID-19 and the impact of

handwashing interventions: The case of Zimbabwe

Short Title: The spatially disaggregated probability of severe COVID-19 and handwashing

interventions in Zimbabwe

Corresponding Author: Sophie Ayling, Ph.D. Candidate

University College London

London, UNITED KINGDOM

Reviewer Comments:

In this article, the authors addressed to estimate of the potential impact of improved access to handwashing on the risk of respiratory illness and its knock-on impact on the risk of developing severe COVID-19 disease across Zimbabwe. A geospatial model that allows the estimation of differential clinical risk at the district level was considered, in which the ideas is sounding great. Thus, the authors need to consider the following minor concerns, and address them point by point prior they move for publication:

• The abstract needs to clearly address the specific spatial method suggested to achieve the proposed objectives.

• The contribution of the work is not clear as compared with the art of the works. For instance, it’s better if the contribution of the latest work compared with the spatial-based work like (Mu X, Yeh AG-O, Zhang X (2021) and etc) is briefly explained.

• I suggest considering references that are more relevant to the current work related to spatial and cast out all which are not relevant.

• The discussion lacks references see the first paragraph of your discussion subsection.

• The datasets were taken from different sources, thus it’s better if the authors put support with the sources. The issue of the quality of the data needs to be well explained in a separate paragraph of the data source. Under the methodology the authors mentioned as they calculated the prevalence of risk factors and 158 comorbidities for severe COVID-19 at the district level, the steps how the calculation was made are not clearly addressed.

• The mathematical writing up of the entire formulas considered is not well written and clearly defining each and every term is also good. The methodology part clearly describing the models and formulas.

• Bring figures and describe them in the body of the manuscript before the references.

• The discussion part also needs to be well described as compared with the baselines. For instance, the first paragraph under the subsection should be compared with some state of the arts.

• Under the conclusion/discussion the authors mentioned as there is a relationship between access to handwashing facilities and respiratory illness, a 325 known risk factor for severe COVID-19.

• Justifying the existence of the spatial variation in the risk of severe COVID-19 needs to take into consideration.

• Better if the authors explain the spatial components taken into account in the latest work. For instance, the results, concluded as there are spatial differences in the risk of severe COVID-19. This needs to be verified through inferential spatial analysis.

Conclusion: I recommend the editor to consider the article for publication just after incorporating my major concerns.

6. PLOS authors have the option to publish the peer review history of their article (what does this mean?). If published, this will include your full peer review and any attached files.

Reviewer #1: No

Reviewer #2: **Yes: **Habte Tadesse Likassa

---

## [Author Response · Author response to Decision Letter 0]

22 Aug 2023

Dear Reviewers and Editorial Team, 

Thank you for taking the time to read and review our manuscript “Estimating spatially disaggregated probability of severe COVID-19 and the impact of handwashing interventions: The case of Zimbabwe”. We greatly appreciate the thoughtful, constructive, and detailed feedback. We have worked to address all of the comments received. Please see the document "Response to Reviewers Letter Form_160823" for our responses.

Thank you for your consideration and we will be happy to address any additional comments or feedback.

Regards,

Author Team

---

## [Editor Report · Decision Letter 1]

26 Sep 2023

Estimating spatially disaggregated probability of severe COVID-19 and the impact of handwashing interventions: The case of Zimbabwe

PONE-D-22-34494R1

Dear Dr. Ayling,

We’re pleased to inform you that your manuscript has been judged scientifically suitable for publication and will be formally accepted for publication once it meets all outstanding technical requirements.

Kind regards,

Bedilu Alamirie Ejigu, Ph.D

Academic Editor

PLOS ONE
---

## [Editor Report · Acceptance letter]

10 Oct 2023

PONE-D-22-34494R1 

Estimating spatially disaggregated probability of severe COVID-19 and the impact of handwashing interventions: The case of Zimbabwe 

Dear Dr. Ayling:

I'm pleased to inform you that your manuscript has been deemed suitable for publication in PLOS ONE. Congratulations! Your manuscript is now with our production department. 

Kind regards, 

on behalf of

Dr. Bedilu Alamirie Ejigu 

Academic Editor

PLOS ONE